# Effects of 12 Weeks of Hypertrophy Resistance Exercise Training Combined with Collagen Peptide Supplementation on the Skeletal Muscle Proteome in Recreationally Active Men

**DOI:** 10.3390/nu11051072

**Published:** 2019-05-14

**Authors:** Vanessa Oertzen-Hagemann, Marius Kirmse, Britta Eggers, Kathy Pfeiffer, Katrin Marcus, Markus de Marées, Petra Platen

**Affiliations:** 1Department of Sports Medicine and Sports Nutrition, Ruhr University Bochum, 44801 Bochum, Germany; marius.kirmse@rub.de (M.K.); markus.demarees@rub.de (M.d.M.); petra.platen@rub.de (P.P.); 2Medizinisches Proteom-Center, Medical Faculty, Ruhr University Bochum, 44801 Bochum, Germany; britta.eggers@rub.de (B.E.); kathy.pfeiffer@rub.de (K.P.); katrin.marcus@rub.de (K.M.)

**Keywords:** proteomics, proteome, collagen hydrolysate, resistance exercise, protein supplementation

## Abstract

Evidence has shown that protein supplementation following resistance exercise training (RET) helps to further enhance muscle mass and strength. Studies have demonstrated that collagen peptides containing mostly non-essential amino acids increase fat-free mass (FFM) and strength in sarcopenic men. The aim of this study was to investigate whether collagen peptide supplementation in combination with RET influences the protein composition of skeletal muscle. Twenty-five young men (age: 24.2 ± 2.6 years, body mass (BM): 79.6 ± 5.6 kg, height: 185.0 ± 5.0 cm, fat mass (FM): 11.5% ± 3.4%) completed body composition and strength measurements and vastus lateralis biopsies were taken before and after a 12-week training intervention. In a double-blind, randomized design, subjects consumed either 15 g of specific collagen peptides (COL) or a non-caloric placebo (PLA) every day within 60 min after their training session. A full-body hypertrophy workout was completed three times per week and included four exercises using barbells. Muscle proteome analysis was performed by liquid chromatography tandem mass spectrometry (LC-MS/MS). BM and FFM increased significantly in COL compared with PLA, whereas no differences in FM were detected between the two groups. Both groups improved in strength levels, with a slightly higher increase in COL compared with PLA. In COL, 221 higher abundant proteins were identified. In contrast, only 44 proteins were of higher abundance in PLA. In contrast to PLA, the upregulated proteins in COL were mostly associated with the protein metabolism of the contractile fibers. In conclusion, the use of RET in combination with collagen peptide supplementation results in a more pronounced increase in BM, FFM, and muscle strength than RET alone. More proteins were upregulated in the COL intervention most of which were associated with contractile fibers.

## 1. Introduction

Skeletal muscle is a dynamic tissue that adapts to external conditions and reacts to acute and long-term physical activity and feeding conditions [1]. Maintenance of muscle mass is not only important for athletes, but also for health and aging [2]. In particular, resistance training using a repeated high-weight stimulus optimizes the preservation of muscle mass that is correlated with body strength [3]. The underlying mechanism of adaptation is an increase in muscle protein synthesis (MPS) stimulated through mechanical load, which can be further augmented by additive protein supplementation [4]. Amino acids are required for protein synthesis. Leucine, a branched-chain amino acid (BCAA), has been shown to be a prerequisite stimulator of the mammalian target of rapamycin (mTOR) signaling pathway, which plays a critical role in MPS [5,6]. 

Besides the contractile muscle components, passive tissues like bone, cartilage, ligaments, and tendons also adapt to resistance training [7]. In this context, Mackey et al. [8] showed that the synthesis of collagen in fibroblasts behaves similarly to MPS after a single bout of resistance training. Collagen is the most represented protein in the human body (30% of the protein concentration). It is an important element of the extracellular matrix (ECM) of skeletal muscle and tendons and is mainly responsible for their functionality in terms of force transmission, flexibility, and adaptation [9]. Thus, to ensure muscle fiber strength transmission, the ECM structures need to adapt in response to resistance training.

The benefit of collagen supplementation on collagen synthesis regarding tendon architecture has been demonstrated by various studies through the analysis of injured participants [10,11]. Studies have reported less joint pain perception after collagen peptide supplementation in healthy active participants and in patients with osteoarthritis [12,13,14,15,16,17], which could be of interest for those involved in elite sport. Clifford et al. [18] reported a faster recovery from jumping exercise and a tendency to reduce muscle soreness after collagen supplementation. 

The beneficial effects of collagen supplementation on passive structures are associated with higher muscle strength following an improvement in force transmission. A study by Zdzieblik et al. [19] was, to the best of our knowledge, the first study to find a positive effect of collagen supplementation on strength and body composition. In a double-blind controlled trial, 53 elderly men (age: 72.2 ± 4.7 years) completed a 12-week resistance training intervention in combination with supplementation with 15 g of collagen or a placebo, resulting in significantly higher gains in fat-free mass (FFM) and muscle strength in the collagen peptide group. However, the underlying mechanism by which collagen peptides act on muscle protein metabolism remains unclear. Collagen is rich in non-essential amino acids, like proline and glycine, and relatively poor in essential amino acids like leucine, the main trigger for MPS.

Proteomic analysis methods have been increasingly developed and used to provide insight into molecular pathways and analyze responses to physical activity or nutrients [20,21,22,23]. The advantage of proteomic analysis based on mass spectrometry is the large number of proteins identified, which provides information about all processes with no need to select specific pathways.

To obtain novel information about long-term adaptations and changes in protein content in response to collagen hydrolysate supplementation in combination with hypertrophy resistance training, we designed a double-blind, randomized, placebo-controlled trial and analyzed muscle proteome before and after a 12-week intervention. 

## 2. Materials and Methods 

### 2.1. Experimental Design

The purpose of this double-blind, randomized, placebo-controlled trial was to investigate the effects of 12 weeks of hypertrophy resistance training combined with collagen supplementation on the proteome of the skeletal muscle. As this was part of a larger investigation project, other parameters, like immune histochemical or transcriptome analysis, were analyzed but will be the content of future publications [24]. In this trial, participants completed a 12-week progressive hypertrophy training intervention period with three controlled training sessions per week. All supervised training sessions included the following exercises in different orders: squat (SQ), dead lift (DL), bench press (BP), rowing (R) with barbells, and knee extensions. After a standardized warm-up, one set of 10 repetitions at 50% of the participant’s 1-repetition maximum (RM) was performed followed by three sets of 10 repetitions at 70% 1-RM with a 2 min break between sets. 

Before and after each intervention period, which contained at least 32 training sessions over 12 weeks, three standardized test days were completed. The test protocols included anthropometric measurements, isokinetic strength testing, 1-RM-testing, and muscle biopsies. A timeline of the test days including muscle biopsies is shown in Figure 1.

During the intervention period, a three-day self-reported nutrition diary was used to evaluate the nutritional intake of the participants. The documentation included three consecutive days, including one day on the weekend and two days during the week. The diaries were evaluated to assess daily calories, carbohydrate, fat, and protein intake with PRODI^®^ 6.0 (Nutri-Science GmbH, Freiburg, Germany).

### 2.2. Participants

Twenty-five healthy sports students participated in the study. To be included in the study, participants had to have basic knowledge of the technical skills required for barbell use, and they had to reach a squat performance of at least 100% of their own body weight. Participants were randomly assigned either to the collagen peptide (COL) (*n* = 12) or placebo (PLA) (*n* = 13) group. 

The sample size of *n* = 25 (COL: *n* = 12, age: 24.4 ± 2.3 years, body height: 185.8 ± 5.0 cm, body mass: 81.4 ± 6.6 kg; PLA: *n* = 13, age: 23.9 ± 2.9 years, body height: 184.3 ± 5.0 cm, body mass: 77.9 ± 4.1 kg) was used for the experimental trial proteome analysis. Subjects were informed about the study design and possible risks and provided their written informed consent. None of them had any health restrictions that were expected to influence the results. The Ethical Advisory Committee of the sport science faculty of the Ruhr-Universität Bochum approved the study protocol.

### 2.3. Supplementation

The supplement administered to the intervention group (*n* = 12) contained 15 g collagen hydrolysate provided by GELITA AG (Bodybalance™, Eberbach, Germany), whereas the subjects in PLA (*n* = 13) received a non-caloric silicon dioxide supplement. The participants consumed the collagen supplement or placebo daily during the 12-week intervention period. On training days, the supplement was consumed dissolved in 250 mL water under observation immediately after each training session. Participants were instructed not to consume additional caloric foods and beverages within 60 min after finishing training to avoid any cross interactions. On days without training, the supplement was taken at a similar time point to distribute the intake and allow about 24 h between ingestions. 

### 2.4. Test Protocols

Three different test days were absolved in the same order and on the same weekday before and after the intervention period (Figure 1). Following a 12-h fasting period, total body mass (BM), fat-free body mass (FFM), and fat mass (FM) were determined using a bioelectrical impedance analysis system (InBody 770, JP Global Markets GmbH, Eschborn, Germany). After a small self-prepared breakfast, 1-RM testing was performed using the method described by Kreamer et al. [25] for SQ, DL, BP, and R. 

After a familiarization session, leg extension maximal voluntary isometric contraction (MViC) testing was performed using only the right leg with a 60° knee flexion angle on a dynamometer (Isomed 2000, D and R GmbH, Hemau, Germany).

Before the participants completed the last test day, three training sessions were performed at 50% 1-RM to learn the training procedures. The first training session at 70% 1-RM was completed on the third test day combined with muscle biopsies. 

After a 12-h fasting period, participants provided a venous blood sample and then consumed a standardized breakfast provided by the investigators. To detect non-responders, a venous blood sample was again taken 2 h after supplement ingestion to determine the concentration of hydroxyproline, an amino acid that frequently appears in collagen hydrolysate [10,26]. Next, a muscle biopsy was taken standardized percutaneously from the vastus lateralis muscle of the right leg under local anesthesia (Xylocitin^®^ 2% with Epinephrine, mibe GmbH, Brehna, Germany) with a 5 mm Bergstrøm needle, as described previously [27]. Muscle samples were cut into a coherent piece of approximately 30 mg and immediately frozen as native tissue in cooled liquid nitrogen and stored at −80 °C until proteome analysis. Immediately after the muscle biopsies, the participants performed a training session at 70% 1-RM followed by ingestion of the collagen or placebo supplement. Two hours later, another blood sample was collected and a muscle biopsy was performed. The second muscle biopsy was taken 3 cm proximal to the previous cut from the same leg. The proteome analysis was performed with the muscle sample collected before training in the fasted state to analyze longitudinal differences.

### 2.5. Proteomics

Muscle samples (25–50 mg) were pulverized in liquid nitrogen and homogenized on ice and resuspended in urea buffer (7 M urea, 2 M thiourea, 20 mM Trisbase pH 8.5) to produce an accurate lysis. Resuspended samples were sonicated 6 times for 10 s each, with 10 s rest on ice to support the lysis process. The protein concentration was determined using the Bradford assay. 

In solution, digestion with trypsin was conducted as described in Winter et al. [28]. Briefly, samples were reduced with dithiothreitol (DTT) for 20 min at 56 °C and alkylated at room temperature (RT) using iodoacetamide (IAA). Trypsin was added in a sample/enzyme ratio of 1:40. Digestion occurred overnight at 37 °C and was stopped by acidification. The peptide concentration was determined by amino acid analysis.

We used 200 ng of peptides from each sample for label-free quantitative liquid chromatography-mass spectrometry (LC-MS) analysis. Liquid chromatography was performed on an UltiMate 3000 RSLC nano LC system (Dionex, Idstein, Germany), as described in Winter et al. [28]. The high-performance liquid chromatography (HPLC) system was online-coupled to the nano electrospray ionization (ESI) source of a Q Exactive mass spectrometer (Thermo Fisher Scientific, Schwerte, Germany). In the ESI-tandem mass spectrometry (MS/MS) analysis, full MS spectra and MS/MS scans were recorded, as described in Chen et al. [29]. 

Progenesis Software (Progenesis QI for Proteomics, Nonlinear Dynamics Ltd., Newcastle upon Tyne, UK) was used for the quantification of peptides. The data were analyzed as described in Winter et al. [28]. Spectra were identified by the Mascot search engine (version 2.5, Matrixscience, London, UK) using the SwissProt part of the UniProtKB [30] for homo sapiens (release 2017_6, containing 20,206 entries and decoys). Decoys were generated by the DecoyDatabaseBuilder for each protein [31]. The mass tolerance was set to 5 ppm for precursor ions and 20 mmu for fragment ions. Due to the sample preparation steps, carbamidomethylation of cysteine was set as a fixed modification, and oxidation of methionine was set as a variable modification. Two tryptic miscleavages were considered in the analysis. The peptide identifications were imported into Progenesis (Nonlinear Dynamics Ltd., Newcastle upon Tyne, U.K.) and assigned to the respective features. Progenesis uses the principle of protein grouping, when one peptide cannot unambiguously be assigned to one single protein, but is unique for a group of proteins. In the case of protein grouping, only the first accession was used for protein assignment and for the evaluation of the results. Nevertheless, all assigned accessions are reported in the respective tables and materials. 

False discovery rate (FDR) was estimated separately for each search using the Protein Inference Algorithm (PIA) [32,33,34]. Here, the Mascot Ion Score thresholds for 1% FDR were determined and taken as cut off values for peptide identification. Proteins quantified with an ANOVA *p* value ≤ 0.05 were considered to be significantly differently expressed and were used for further evaluation. 

### 2.6. Statistical Analysis

Data are presented as means and standard deviations. Statistical analyses were performed using the statistical program SPSS Statistics 25 (IBM, Armonk, New York, USA). The Kolmogorov–Smirnoff test was used to test the normal distribution of variables (differences). A two-way ANOVA with repeated measurements (time × treatment) was used to analyze significant differences between groups over time. If the ANOVA showed a significant effect (*p* ≤ 0.05), post-hoc tests were performed using paired *t*-tests (pre vs. post), and *t*-tests were used for independent samples to detect differences between groups. The level of statistical significance was set at *p* ≤ 0.05 and this was alpha-adjusted (Bonferroni correction). 

Cut-off values for proteins that were significantly regulated between the two groups were determined by a fold change (FC) of at least 1.5 and a *q* value ≤ 0.05 (corrected ANOVA *p* value in accordance with Benjamini Hochberg) [35,36]. For further analysis, PANTHER (http://www.pantherdb.org/) (University of Southern California, Los Angeles, CA, USA) free analysis tools were used for Gene Ontology categorization and pathway analysis [37].

## 3. Results and Discussion

The analysis of the blood samples revealed significant increases in hydroxyproline for each participant in COL two hours after ingesting the collagen peptide (pre: 33.3 ± 19.7 µmol/L, post: 95.8 ± 27.1 µmol/L; *p* ≤ 0.0125) compared to participants in the PLA group (pre: 14.4 ± 6.4 µmol/L, post: 14.3 ± 11.3 µmol/L; *p* = 0.956), indicating that all subjects were able to absorb the hydrolyzed collagen [10]. 

The dietary assessment showed no differences in caloric intake or macronutrients between the groups. The average protein intake over three days including the supplement was 135.1 ± 29.0 g/day in COL and 145.8 ± 52.7 g/day in PLA (*p* = 0.557), representing an adequate and equal protein intake of 1.66 g/kg/day (COL) and 1.86 g/kg/day (PLA). 

At the beginning of the 12-week intervention period, no statistical differences existed between COL and PLA for any anthropometric or strength parameter (Table 1). From pre to post (main effect *p* ≤ 0.05) the whole cohort increased significantly in all parameters. ANOVA with repeated measurements showed significant interaction effects (*p* ≤ 0.05) for BM, FFM, and R, and trends for an interaction effect for SQ (*p* = 0.073), BP (*p* = 0.099), and MViC (*p* = 0.066), which increased slightly more in COL than in PLA, respectively. FM and DL showed no interaction effect between groups. Post-hoc *t*-tests with the Bonferroni correction (to *p* ≤ 0.0125) revealed no significant differences between groups in BM, FFM, and R after 12 weeks. FFM only increased in COL, whereas no difference was found in PLA. 

Considering the differences between groups, delta values from pre to post are given in Table 2. Individuals in COL gained significantly more BM and FFM in contrast with those in PLA. Individuals in COL had a significantly higher strength increase in R compared to those in PLA (Table 2).

The study by Zdzieblik et al. [19] showed a clear improvement in isokinetic quadriceps strength, a greater loss in FM and higher gains in FFM in 26 sarcopenic older men supplementing 15 g collagen peptides per day during a 12-week training intervention compared with a placebo group (*n* = 27). In our study, FFM also increased to a higher extend in COL compared with PLA, while strength only slightly increased further in COL compared with PLA, suggesting that mechanisms are more pronounced in the elderly populations than in younger healthy men. Because collagen has a high impact on tendon tissues [10,11], it is possible that exercises that stimulate reactive power of the tendons would have shown clearer effects of collagen peptide supplementation [18]. The strength tests used in this study only slightly depend on reactive forces and might therefore have underestimated possible effects of the supplement on tendon tissues. Another part of our study analyzed the muscle fiber cross-sectional area and showed no differences between groups [24], underpinning the hypothesis that adaptations induced by collagen supplements do not affect the muscle cell per se but structural components of muscle tissue surrounding the cells.

Overall comparisons of 1377 proteins or protein groups (proteins and protein groups are referred to as “proteins” in the following text for better readability) were quantified in our analysis using Progenesis QI for proteomics. 

The baseline data of the two groups were compared against each other to identify any differences on the proteomic level due to the cohort composition itself, to prevent possible data misinterpretation due to cohort variability. In total, three proteins were significantly increased in PLA compared with COL prior the intervention due to biological variability (Table 3). In COL, no higher abundant proteins could be identified compared with PLA. In order to avoid possible influences of the different baseline values, these proteins were excluded from further data analysis.

The distribution of regulated proteins before and after 12 weeks of resistance training and supplementation differed; there were 17 downregulated proteins in COL compared to 18 downregulated proteins in PLA, of which 9 proteins appeared in both groups (Figure 2a). We found that 293 proteins were highly regulated in COL after the 12-week intervention, of which 72 were also of higher abundance in PLA. These changes thus resulted from the hypertrophy training intervention. Hence, 221 proteins were only upregulated in COL compared to 44 proteins in PLA, showing a pronounced difference in the number of upregulated proteins between the two groups (Figure 2). The direct comparison between COL and PLA after 12 weeks of supplementation and resistance exercise training revealed no significant different proteins. 

For a deeper proteomic characterization of our data set Gene Ontology (GO) categorization was performed using PANTHER (default settings using Fisher’s exact and FDR-correction) in respect to three different aspects: Cellular components, biological processes, and molecular functions [37]. To provide a better overall view of the large number of hits, the REVIGO tool [38] was used to sum classes ranked by FDR (Figure 3a–c). The downregulated proteins (Figure 2a) were not considered because the low number (PLA = 9; COL = 8) of hits was not appropriate for group allocations. 

Higher abundant proteins in the intersection (Figure 2b) were assigned to GO categories often found in response to resistance training. For example, upregulated proteins in the GO category “biological processes” (Figure 3a) included the response to cytokines or protein modifications, which are in line with other study outcomes [39,40]. No subclass of the aspects “molecular functions” and “cellular components” showed specific categories associated with skeletal muscle. Upregulated proteins in PLA showed no direct association with the skeletal muscle system, although the upregulated proteins in PLA in the biological processes category that could be classified into organonitrogen compound metabolic processes and nitrogen compound metabolic processes indicated some responses to an amino acid stimulus (Figure 3c).

In contrast, annotation of upregulated proteins of COL included those associated with GO biological processes related to the skeletal muscle (muscle contraction and muscle system process) and protein modifications (protein ubiquitination, protein modification by small protein conjugation or removal) (Figure 3b). 

The annotated GOs for cellular components were similar in COL and PLA: extracellular parts, cytoplasm, intracellular, and vesicles. However, PLA contained a higher number of upregulated proteins associated with the terms mitochondria and mitochondrial matrix (Figure 3c), and COL had a few upregulated proteins related to the GO category supramolecular fibers, which are superior to contractile fibers (Figure 3b). 

The GOs belonging to molecular function in COL also included categories directly related to the skeletal muscle fiber structure, like cytoskeletal protein binding and the structural constitution of muscle. Upregulated proteins in PLA were assigned with general categories, such as catalytic activity and coenzyme binding, with respect to their molecular function.

Overall, COL showed a strong association with annotations spanning contractile muscle fiber (sarcomere or cytoskeleton) in all three GO categories, whereas PLA showed none of them or not enough regulated proteins to form a category. To summarize, we observed considerable differences in the protein response due to collagen supplementation, as both groups performed the same training protocol. 

A pathway analysis was performed using PANTHERs overrepresentation test with default settings and the “Reactome pathways” annotated data set. The main categories of significant regulated pathways (Cut off values: FDR ≤ 0.01, fold enrichment score ≥ 2.00 and at least three or more proteins assigned) are presented in Figure 4. Most pathways were annotated to cell cycle, signal transduction, metabolism of proteins and immune system. Selected pathways that are relevant in the context of exercise training and adaptation according to the literature are listed in Table 3 (complete list in Appendix A). COL showed 192 pathways that are significantly overrepresented. In comparison, only one pathway was assigned to PLA according to the cut-off criteria, which was “metabolism” (fold enrichment = 3.70; FDR = 4.75 × 10^−3^).

Overrepresented pathways in COL showed many responses characteristic to exercise training and adaptation (Table 4). Of 192 significant pathways, 31 pathways were assigned to signal transduction. Several mitogen-activated protein kinases (MAPK) pathways were overrepresented that have been reported before in human skeletal muscle, after cycling or running exercise [41,42]. MAPK cascades are activated through cellular stress such as muscle contraction and exercise. In particular, MAPK1 and MAPK3, also known as extracellular-signal regulated kinase (ERK) 2 and 1, are the key components of the MAPK/ERK cascade, which regulates cell growth and differentiation through influencing transcription, translation, and cytoskeleton structure [43]. Furthermore, signal transduction through protein kinase B (Akt) is upregulated in COL. The Akt/mTOR pathway is one of the most researched pathways on cell growth and is regarded as the main regulator of hypertrophy [44]. As expected after exercise, the immune system response increased cytokine signaling through interleukins to improve the regeneration process [45]. The metabolism of proteins belongs mainly to translational processes and protein folding, especially for cytoskeletal proteins. Overrepresented metabolic pathways are mainly glycolytic energy metabolism associated pathways as expected according to the chosen training stimulus [46]. 

Cellular response to external stimuli pathway, showed downstream cascades responding to stress, hypoxia and heat stress. Hypoxia inducible factor alpha 1 (HIF-1) is regulated through muscle oxygen concentration that decreased during exercise or injury. HIF-1 stimulates via erythropoietin (EPO) myoblast proliferation and muscle recovery after injury and is important for muscle repair and regeneration [47]. Furthermore, heat shock transcription factor 1 (HSF1) is present in multiple pathways. HSF1 acts as the major transcription factor in human muscle cells in response to stress. Striated and smooth muscle contraction pathways are also significantly overrepresented after 12 weeks of strength training and collagen supplementation.

To create an initial understanding of the related mechanisms, proteins belonging to the term contractile fibers (assigned according to PANTHER) were further analyzed, since proteins of this category were detected in both groups. Only two proteins from 44 differentially upregulated proteins of PLA were assigned to this type, in COL 26 out of 221 higher abundant proteins were annotated to the term contractile fibers (Table 5 and Appendix A). One of the upregulated proteins in PLA was alpha-crystallin B chain, which often increases to protect the cytoskeleton and myofibril structure after muscle damage caused by eccentric exercise [48]. The other protein belonging to this term detected in PLA was leiomodin-2, which promotes the polymerization of actin filaments in skeletal muscle cells [49]. 

In COL several proteins with different myofibrillar function were identified as being in higher abundance compared to PLA, e.g., myosin proteins (myosin regulatory light chain 2, myosin-8, myosin light chain 1/3) and actin-binding proteins (alpha-actinin-2, synaptopodin, tropomodulin-4), including tropomyosins (tropomyosin beta chain, tropomyosin alpha-3 chain). Many have already been observed before in proteome analyses after resistance exercise [50]. Troponin C and I were also found to be of higher abundance in COL. Troponin I is already known to be increased by chronic exercise and regulates the interaction between actin and myosin filaments [51]. One of the upregulated proteins belonging to the Z-disk (five proteins total) is myotilin, an important marker for remodeling the myofibril structures after exercise. Myotilin binds alpha-actinin-2 and PDZ and LIM domain protein 3 to control Z-disk organization [52]. Five sarcolemmal proteins showed an increased fold enrichment, including desmin, which has already been reported in other studies to be a responder to resistance exercise, which is relevant for force transmission through the cytoskeleton [28]. The cytoskeleton is not just a structural component but is also responsible for the communication between the muscle cells and the ECM [9]. Small muscular protein and smoothelin-like protein 1 were also upregulated in COL. These two proteins are associated with the costamere, being responsible for the force transmission, which is generated in the myofibrils and transferred to the ECM [53]. 

COL induced a higher abundance in proteins being associated with resistance training adaptations. These effects might be caused by the high hydroxyproline-peptide content of the collagen peptide supplementation, as Kitakaze et al. [54] were able to show an increase myoblast differentiation and myotube hypertrophy in murine skeletal muscle C2C12 cells by hydroxyproline-glycine-peptides.

The number of proteins belonging to relevant categories of the skeletal muscle (manually annotated) was significantly different between both groups, resulting in more relevant proteins for adaptation processes observed in COL compared with PLA. 

Because participants in COL ingested 15 g collagen hydrolysate every day, collagen proteins and collagen-associated proteins that were differently expressed after 12 weeks are shown in Table 6. Collagen 6alpha2, fibronectin, and laminin subunit gamma 1 showed increased FC in both groups after 12 weeks as a result of resistance training. Laminins, collagens, as well as fibronectins are the main fibrous parts of the ECM. Fibronectin has many different functions, for example, organization of ECM structures, cell migration, and adhesion and tissue repair through fibroblast migration [55]. The increase in fibronectin after resistance training is consistent with other studies that reported increased values 72 h after exercise [56]. An increase in laminin was associated with myogenic differentiation in mice [57]. Consequently, both groups adapted their ECM protein production independently from collagen supplementation. 

In contrast, collagen 5alpha1, collagen 15alpha1 and collagen 18alpha1 were only upregulated in COL, whereas PLA showed a fold enrichment of the protein lumican. Collagen 5alpha1 is a fibrillar collagen normally contained in collagen 1-rich regions, playing a critical role in fibrillogenesis and fibril organization [58]. It is mostly associated with Ehlers-Danlos syndrome, which is caused by a congenital mutation of the *col5a1* gene and is characterized by hyperextensibility of skin and joints [59]. Baghdadi et al. [60] identified collagen 5alpha1 in muscle stem cells as a regulator of the notch signaling pathway that is concerned with the prevention of satellite cell storage by repressing differentiation process initiated by myoD (myogenic differentiation 1) to maintain quiescence, in addition to the main function of organizing the structure of the ECM through collagen 1 formation. The FC of collagen 5alpha1, which is extremely high compared with all other proteins, should be considered with caution because data validation (using immunoblotting) is still missing, and proteomic analysis itself struggles with a high variability between individuals [61,62].

The other protein belonging to the collagen family that was upregulated in COL is collagen 18alpha1. Together with collagen 15, it forms a collagen category named multiplexins, which is characterized by compounds with multiple triple-helix domains in containing non-collagenous parts. Collagen 18alpha1 is located next to collagen 4 in the basement membrane, occurs nearly ubiquitously in the human body, and seems to differ in function depending on the location [63]. It contains a non-collagen domain called endostatin, which is known to be a suppressor of angiogenesis and tumor growth [64]. Because of its inhibitory effect on tumor growth, endostatin has been well investigated in contrast to the function of collagen 18alpha1 in different tissues. Its role in skeletal muscle has not been directly investigated, but it is transferred from other tissues where it is the structural component of the basement membrane, binding ECM proteins like laminin. 

Lumican, which was upregulated in PLA, is also a component of the ECM that binds to collagen 1 and affects collagen fibrillogenesis and therefore, the matrix assembly [65]. 

Connective tissue remodeling has been studied in previous years, as *collagen 1, 3*, and *4* genes are elevated after a single bout of exercise, but are unchanged after a second bout [66]. After 36 training sessions, we would have expected a higher abundance in these proteins caused by adaptation, but they were unaffected in both groups, and instead, other collagenous proteins were upregulated in both groups. This shows their importance not only for contractile fibers but also for passive structures, like the ECM around muscle cells, to adapt to the mechanical load [9]. The adaptation of the ECM seems to occur even without collagen supplementation, although more collagenous proteins were upregulated following COL supplementation. 

## 4. Conclusions

In conclusion, after 12 weeks of hypertrophy resistance exercise training in combination with collagen supplementation compared with a placebo, we found significantly higher BM and FFM and a slightly more pronounced increase in strength in COL compared with PLA. Individuals in COL showed a significantly higher number of upregulated proteins and significantly more pathways associated with resistance exercise compared with PLA after the intervention, indicating stronger effects for the combination of strength training and supplementation on the skeletal muscle proteome than strength training alone. In contrast to these considerable effects on the proteome, increments in muscle strength were not as clear. Further investigations are needed to clarify the role of collagen 5alpha1 in muscle adaptations to resistance training and collagen supplementation and its possible impact on strength. Furthermore, it would be interesting to compare different protein sources with regard to their metabolic effects and strength training adaptations.

## Figures and Tables

**Figure 1 nutrients-11-01072-f001:**
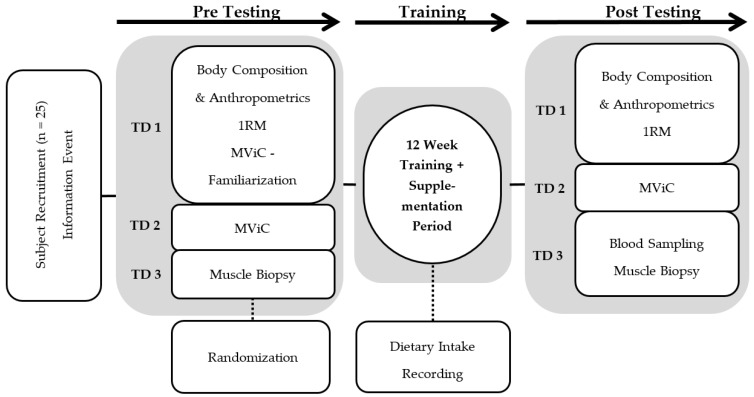
Flow chart describing the experimental design. Pre- and post-testing consisted of three test days (TD1–3) separated by a 12-week intervention period with collagen or placebo supplementation each day and hypertrophy resistance training three days per week. MViC: Maximal voluntary isometric contraction; RM: Repetition Maximum.

**Figure 2 nutrients-11-01072-f002:**
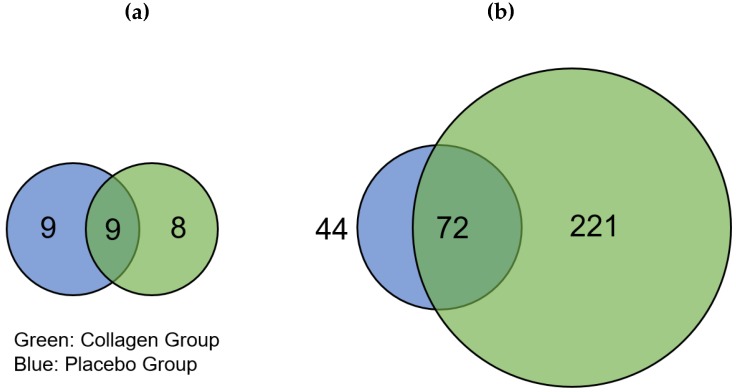
Venn diagrams showing different numbers of (**a**) significantly downregulated and (**b**) upregulated proteins in the placebo (blue) and collagen (green) groups after 12-week resistance exercise training and supplementation. The intersection represents proteins that were regulated in both groups.

**Figure 3 nutrients-11-01072-f003:**
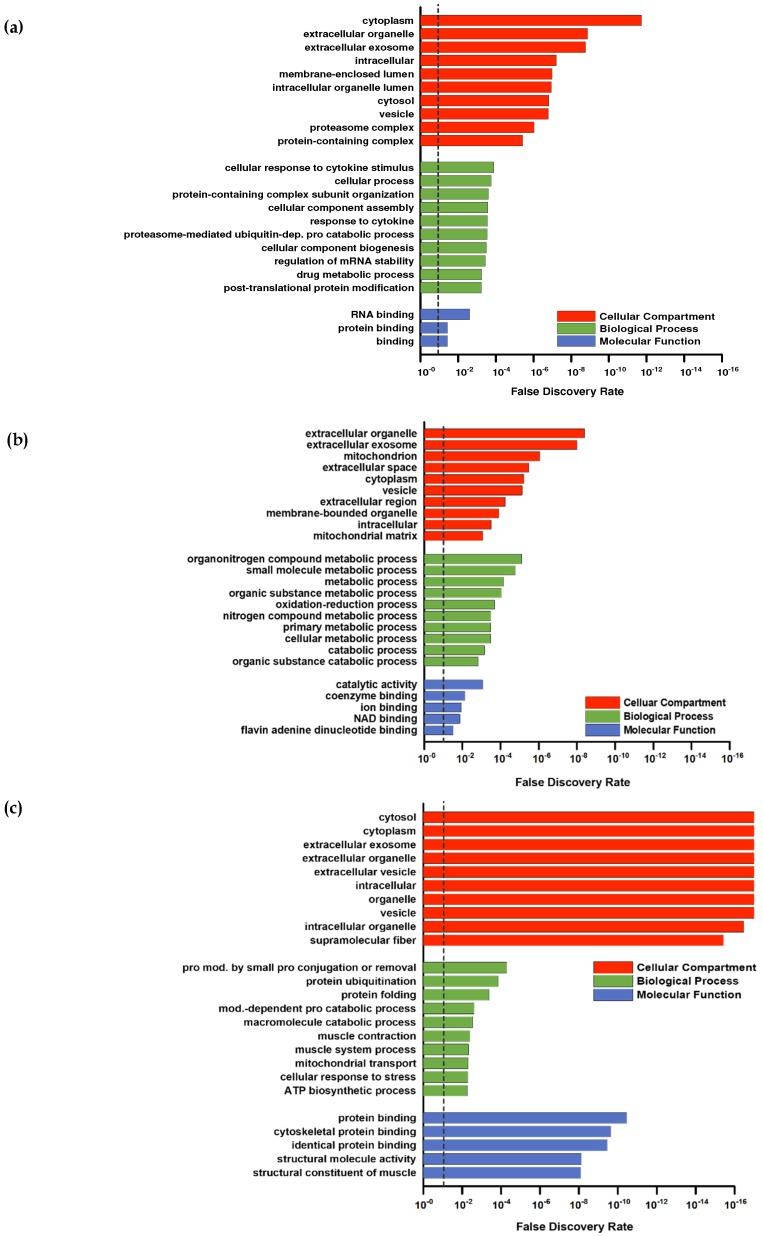
Gene Ontology (GO) analysis of proteins from the (**a**) intersection, (**b**) collagen group, and (**c**) placebo group after 12 weeks of resistance training and supplementation calculated by PANTHER. The false discovery rate (FDR) is marked with a dashed line at 0.10.

**Figure 4 nutrients-11-01072-f004:**
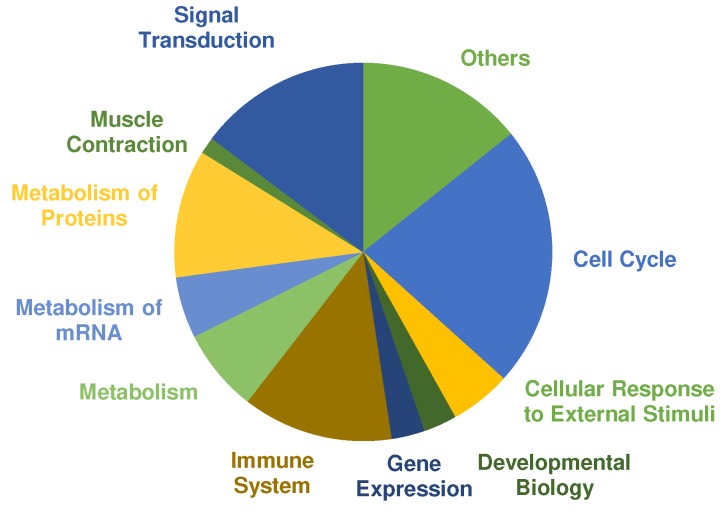
Pie chart shows significantly overrepresented pathways in the collagen (COL) group after 12 weeks of resistance training and supplementation analyzed by PANTHER. Size of the different colored parts indicates number of assigned pathways.

**Table 1 nutrients-11-01072-t001:** Results for body mass (BM), fat mass (FM), fat free mass (FFM), the squat test (SQ), dead lift (DL), bench press (BP), rowing (R), and isometric strength testing (MViC) before and after the 12-week intervention period. Data are presented for the collagen and placebo groups (means ± SD) as a pre- and post-test comparison. Statistical significance for interactions as determined by ANOVA was considered as *p* ≤ 0.05, and the level of significance for the post-hoc *t*-tests was set at *p* ≤ 0.0125 after Bonferroni correction.

Characteristic	Collagen Group (*n* = 12)		Placebo Group (*n* = 13)		Interaction	Post-Test Difference between Groups
	Pre	Post	Pre	Post	*p* Value	*p* Value
BM (kg)	81.4 ± 6.6	84.4 ± 6.3 *	77.9 ± 4.1	79.4 ± 5.1 *	0.035	n.s. (0.039)
FM (kg)	10.3 ± 3.6	10.9 ± 4.1 *	8.4 ± 2.2	9.4 ± 2.4 *	n.s. (0.534)	-
FFM (kg)	71.2 ± 5.7	73.8 ± 5.3 *	69.6 ± 4.0	70.3 ± 4.3 *	0.014	n.s. (0.080)
SQ (kg)	114.3 ± 20.2	140.8 ± 24.3 *	108.7 ± 8.3	126.3 ± 13.9 *	n.s. (0.073)	n.s. (0.084)
DL (kg)	131.7 ± 19.6	156.3 ± 21.2 *	128.0 ± 15.6	143.9 ± 10.6 *	n.s. (0.234)	-
BP (kg)	80.4 ± 14.0	94.4 ± 15.6 *	84.4 ± 13.9	94.2 ± 10.2 *	n.s. (0.099)	n.s. (0.987)
R (kg)	85.0 ± 11.3	98.3 ± 12.6 *	91.2 ± 8.9	97.3 ± 8.1 *	0.025	n.s. (0.812)
MViC (nM)	294.3 ± 56.3	323.6 ± 72.1 *	260.7 ± 25.0	275.2 ± 31.3 *	n.s. (0.066)	n.s. (0.059)

* significantly different to pre-test (post-hoc paired *t*-test with Bonferroni correction to *p* ≤ 0.0125). n.s. = not significant.

**Table 2 nutrients-11-01072-t002:** Differences (mean ± SD) in post-test values between the collagen (COL) group and placebo (PLA) group in body mass (BM), fat mass (FM), fat free mass (FFM), the squat test (SQ), dead lift (DL), bench press (BP), rowing (R), and isometric strength testing (MViC) before and after the 12-week intervention period. Unpaired *t*-tests were used to detect differences between both groups after the intervention (level of significance was set to *p* ≤ 0.05).

Exercise	Collagen Group	Placebo Group	Unpaired *t*-test
	Delta	Delta	*p* value
BM (kg)	3.01 ± 2.01	1.50 ± 1.29	0.035
FM (kg)	0.58 ± 1.47	0.90 ± 1.09	n.s. (0.534)
FFM (kg)	2.56 ± 2.22	0.70 ± 1.14	0.014
SQ (kg)	26.5 ± 13.2	17.6 ± 9.9	n.s. (0.073)
DL (kg)	24.5 ± 19.1	15.9 ± 13.9	n.s. (0.234)
BP (kg)	14.0 ± 6.6	9.8 ± 5.5	n.s. (0.099)
R (kg)	13.3 ± 8.7	6.1 ± 5.8	0.025
MViC (nM)	29.4 ± 22.9	14.5 ± 14.3	n.s. (0.066)

n.s. = not significant.

**Table 3 nutrients-11-01072-t003:** Upregulated proteins in the collagen (COL) and placebo (PLA) group showing baseline differences before the intervention.

UniProt	Gene Name	Protein Description		Fold Change (FC)	
				Collagen	Placebo
Only upregulated in COL before 12 weeks					
	-			-	-
Only upregulated in PLA before 12 weeks					
P10636	MAPT	Microtubule-associated protein tau		-	1.63
P02144	MB	Myoglobin		-	1.71
P09493	TPM1	Tropomyosin alpha-1 chain		-	1.52

**Table 4 nutrients-11-01072-t004:** Overrepresented pathways in the collagen (COL) group after 12 weeks of resistance training and supplementation. Main categories are shown with the number of annotated pathways and downstream pathways with the calculated fold enrichment and false discovery rate (FDR).

Pathway	Number of Pathways	Fold Enrichment	FDR
**Cell Cycle**	**47**		
**Signal Transduction**	**31**		
PIP3 activates AKT signaling		3.98	3.56 × 10^−3^
MAPK family signaling cascades		5.22	7.42 × 10^−6^
MAPK6/MAPK4 signaling		9.65	1.47 × 10^−5^
MAPK1/MAPK3 signaling		6.09	1.63 × 10^−6^
RAF/MAP kinase cascade		6.22	1.37 × 10^−6^
Regulation of RAS by GAPs		12.63	2.48 × 10^−6^
**Immune System**	**27**		
MAP kinase activation		7.57	7.58 × 10^−3^
Cytokine signaling in immune system		3.71	7.00 × 10^−7^
Signaling by Interleukins		4.25	2.51 × 10^−6^
Interleukin-1 family signaling		7.66	8.30 × 10^−6^
Interleukin-1 signaling		10.39	8.47 × 10^−7^
MAP3K8 (TPL2)-dependent MAPK1/3 activation		17.89	9.74 × 10^−3^
Interleukin-12 family signaling		6.70	3.23 × 10^−2^
**Metabolism of Proteins**	**23**		
Translation		3.91	1.13 × 10^−3^
Eukaryotic translation elongation		9.14	2.11 × 10^−5^
Eukaryotic translation initiation		6.42	7.41 × 10^−4^
Eukaryotic translation termination		6.16	6.39 × 10^−3^
Protein folding		7.63	2.41 × 10^−4^
Post-chaperonin tubulin folding pathway		17.35	1.79 × 10^−3^
Cooperation of prefoldin and TriC/CCT in actin and tubulin folding		17.89	3.86 × 10^−5^
Formation of tubulin folding intermediates by CCT/TriC		19.09	1.90 × 10^−4^
Prefoldin mediated transfer of substrate to CCT/TriC		21.21	1.74 × 10^−5^
Post-translational protein modification		2.61	1.89 × 10^−6^
UCH proteinases		11.93	1.04 × 10^−7^
Deubiquitination		5.08	1.00 × 10^−5^
**Metabolism**	**15**		
The citric acid (TCA) cycle and respiratory electron transport		8.27	1.08 × 10^−7^
Pyruvate metabolism and citric acid (TCA) cycle		8.84	4.29 × 10^−3^
Pyruvate metabolism		12.72	4.76 × 10^−3^
Respiratory electron transport, ATP synthesis by chemiosmotic coupling, and heat production by uncoupling proteins		7.76	2.13 × 10^−5^
Respiratory electron transport		7.63	2.42 × 10^−4^
Metabolism of amino acids and derivatives		6.8	4.19 × 10^−11^
Histidine, lysine, phenylalanine, tyrosine, proline and tryptophan catabolism		10.37	2.29 × 10^−3^
Glyoxylate metabolism and glycine degradation		12.31	5.25 × 10^−3^
Gluconeogenesis		11.57	6.32 × 10^−3^
**Cellular Response to external stimuli**	**11**		
Cellular responses to stress		6.31	1.04 × 10^−10^
Cellular response to hypoxia		12.72	7.28 × 10^−7^
Oxygen-dependent proline hydroxylation of HIF-1α		14.46	3.37 × 10^−7^
Cellular response to heat stress		13.16	5.52 × 10^−8^
HSF1 activation		39.77	1.15 × 10^−5^
HSF1-dependent transactivation		31.81	9.12 × 10^−8^
Attenuation phase		34.08	1.98 × 10^−5^
Regulation of HSF1-mediated heat shock response		7.12	9.56 × 10^−3^
HSP90 chaperone cycle for steroid hormone receptors (SHR)		13.63	6.36 × 10^−6^
**Metabolism of mRNA**	**11**		
**Developmental Biology**	**6**		
**Gene Expression**	**6**		
**Muscle Contraction**	**3**		
Striated muscle contraction		23.86	8.26 × 10^−8^
Smooth muscle contraction		10.91	7.55 × 10^−3^
**Others**	**12**		
CREB phosphorylation through the activation of CaMKII		19.09	5.28 × 10^−3^

**Table 5 nutrients-11-01072-t005:** Upregulated proteins belonging to the GO category “contractile fibers” in the collagen (COL) and placebo (PLA) groups.

UniProt	Gene Name	Protein Description	Fold Change	FDR
Only upregulated in COL after 12 weeks of training				
P35609	ACTN2	Alpha-actinin-2	1.57	0.006
Q9GZV1	ANKR2	Ankyrin repeat domain-containing protein 2	1.62	0.025
O95817	BAG3	BAG family molecular chaperone regulator 3	1.87	0.004
O00499	BIN1	Myc box-dependent-interacting protein 1	1.79	0.000
P17661; P07197	DESM	Desmin	1.70	0.002
O00757	F16P2	Fructose-1,6-bisphosphatase isozyme 2	1.51	0.020
Q96A32	MLRS	Myosin regulatory light chain 2, skeletal muscle isoform	1.70	0.007
P13535	MYH8	Myosin-8	2.55	0.047
P05976; Q15111	MYL1	Myosin light chain 1/3, skeletal muscle isoform	1.61	0.009
Q9UBF9	MYOTI	Myotilin	1.51	0.008
O14974	MYPT1	Protein phosphatase 1 regulatory subunit 12A	1.54	0.021
Q53GG5	PDLI3	PDZ and LIM domain protein 3	1.93	0.000
P60900	PSA6	Proteasome subunit alpha type-6	1.56	0.017
Q9UHP9	SMPX	Small muscular protein	2.03	0.000
A8MU46	SMTL1	Smoothelin-like protein 1	2.05	0.003
P23327	SRCH	Sarcoplasmic reticulum histidine-rich calcium-binding protein	1.96	0.003
Q9H7C4	SYNCI	Syncoilin	1.51	0.026
O15061	SYNEM	Synemin	1.57	0.003
Q8N3V7	SYNPO	Synaptopodin	1.82	0.016
Q9NZQ9	TMOD4	Tropomodulin-4	1.92	0.002
P02585; P27482	TNNC2	Troponin C, skeletal muscle	1.98	0.002
P48788	TNNI2	Troponin I, fast skeletal muscle	1.98	0.006
P07951; O75330	TPM2	Tropomyosin beta chain	1.88	0.000
P06753	TPM3	Tropomyosin alpha-3 chain	2.07	0.004
Q9BYV2	TRI54	Tripartite motif-containing protein 54	2.15	0.033
Only upregulated in PLA group after 12 weeks of training				
P02511	CRYAB	Alpha-crystallin B chain	2.01	0.037
Q6P5Q4	LMOD2	Leiomodin-2	2.80	0.014

**Table 6 nutrients-11-01072-t006:** Upregulated proteins from the intersection, collagen (COL), and placebo (PLA) groups belonging to collagen proteins or collagen-associated proteins.

UniProt	Gene Name	Protein Description	Fold Change (FC)			FDR	
			Collagen	Placebo		Collagen	Placebo
Upregulated in both groups after 12 weeks							
P12110	CO6A2	Collagen alpha-2(VI) chain	2.86	3.17		0.001	0.000
P02751	FINC	Fibronectin	4.46	2.24		0.007	0.021
P11047	LAMC1	Laminin subunit gamma 1	1.90	1.71		0.023	0.022
Only upregulated in COL after 12 weeks							
P20908	CO5A1	Collagen alpha-1(V) chain	108.74	-		0.018	-
P39059	COFA1	Collagen alpha-1(XV) chain	1.69	-		0.028	-
P39060	COIA1	Collagen alpha-1(XVIII) chain	2.85	-		0.001	-
Only upregulated in PLA after 12 weeks							
P51884	LUM	Lumican	-	1.59		-	0.001

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
