# Peer review of "Effects of 12 Weeks of Hypertrophy Resistance Exercise Training Combined with Collagen Peptide Supplementation on the Skeletal Muscle Proteome in Recreationally Active Men"

_nutrients, 2019, doi:10.3390/nu11051072_

Reviewer 1 Report

The manuscript by Oertzen-Hagemann et al uses a mixed experiment design to investigate the effects of collagen supplementation on the muscle proteome response to 12 weeks of resistance exercise training in 2 independent groups of young healthy males. Participants and experimenters were blind to the randomised allocation of control (placebo) and collagen supplementation. Proteome profiling was conducted using bottom-up label-free quantitation of tryptic peptide mixtures from whole muscle homogenates. Anthropometric and physiological outcomes were also assessed, including 1-repetition maximum (1-RM) and maximal voluntary contraction (MVC). 

Collagen supplementation increased (significant interaction effect) the gains in MVC and 1-RM of back squat and bench-press exercises compared to resistance exercise and placebo control. This is consistent with previous works investigating the effects of collagen supplementation on muscle adaptation to resistance exercise. The novel aspect of the work is the proteomic analysis, which highlights numerous changes in protein abundance that were specific to the response to resistance exercise either with or without collagen supplementation. The muscle proteome response to resistance exercise was greater with collagen supplementation. However, collagen supplementation did not appear to have a focused effect on extracellular matrix proteins.

There are relatively few proteomic studies of resistance exercise training so the current data could be an important contribution to the field. In particular, questions regarding the adaptation of muscle connective tissue is a current area of interest, and the present work provides new insight to the effects of collagen supplementation. 

My major concern with the manuscript relates to the statistical analysis and format chosen for reporting proteomic data. The method section states 2-way ANOVA was used, which is appropriate and is evident in the reporting of physiological data. However, there is no information regarding main or interaction effects of resistance exercise or collagen supplementation in the proteomic data. It is not clear whether differences existed in the muscle proteome between groups before the intervention. Nor is it clear whether the reported proteins that are described as “only up-regulated in the collagen group” represent significant interactions. Data presented in tables 3 and 4 report only fold change and do not provide information on the statistical significance or mean ± sd of each group (these features can be important in the future use of data by others, e.g. for meta-analysis etc.) The results section uses a format more typical of gene expression studies that does not convey proteomic data appropriately or optimally. It is relatively uninformative to report such extensive gene ontological analysis for muscle proteome data. Many of the tests used to denote enrichment assume a comparison against a whole-genome background and so are not appropriate for muscle proteomic analysis that does not include data all gene products. Indeed the results sections lacks information regarding the total number of proteins that were included in the current analysis. Moreover, summarising proteins by gene ontology provides limited biological insight compared to that which can be achieved by considering the literature and biochemical attributes of proteins, interaction networks and metabolic pathways, for example.

The manuscript may also be improved by considering the following minor issues. Is the choice of experiment deign a limitation? From the data presented it is not clear whether there were baseline differences across the muscle proteome of the 2 independent groups of participants. Given the biological variability of human participants, a cross-over design may have been more appropriate/ effective in detecting supplementation-specific effects. What is the reproducibility of the anthropometric and physiological outcome data? Some of the data in Tables 1 and 2 seem quite broadly distributed. Does your data include lower and higher responders and could the data be illustrated to show the individual responses of participants? Can more insight be extracted from the data by investigating correlations between the muscle proteome and physiological data?

Notes/ short questions:

Was the biopsy site standardised?

Which of the 2 biopsy samples (pre- or post-warm-up) was used for comparing longitudinal differences in response to 12-weeks training?

Usually necessary to provide a reference number to confirm appropriate ethic review 

Author Response

Response to Reviewer 1 Comments:

First, I would like to thank you very much for reading my manuscript and for the many constructive comments! I hope that I was answering most of the questions and comments or that I was able to work on them appropriately!

Thank you very much!

 Point 1: My major concern with the manuscript relates to the statistical analysis and format chosen for reporting proteomic data. The method section states 2-way ANOVA was used, which is appropriate and is evident in the reporting of physiological data. However, there is no information regarding main or interaction effects of resistance exercise or collagen supplementation in the proteomic data.It is not clear whether differences existed in the muscle proteome between groups before the intervention.

Nor is it clear whether the reported proteins that are described as “only up-regulated in the collagen group” represent significant interactions. Data presented in tables 3 and 4 report only fold change and do not provide information on the statistical significance or mean ± sd of each group (these features can be important in the future use of data by others, e.g. for meta-analysis etc).

Response 1: For our proteomic analysis a standardized label free quantification approach using Progensis QI for Proteomics was carried out. Here all runs are aligned to each other for optimal replicate comparison. Further on peak picking is performed on all feature maps An aggregated map is then applied to each sample, resulting in 100% matching of peaks with no missing values. All abundances provided are normalized abundances and do not display an absolute value e.g. a concentration. By comparing both groups against each other, our analysis does provide an information about relative changesbetween the two sample groups. Therefore, we added the corrected statistical significance to the p-values in all Tables. To show more interaction between physiological responses and proteins, we performed a pathways analysis through PANTHER (line 308-, Table 4).

We also added baseline differences marked in the text from line 237-245 and Table 3. Where we identified 3 differentially regulated proteins before the intervention, these proteins were excluded from further analysis.

Point 2: The results section uses a format more typical of gene expression studies that does not convey proteomic data appropriately or optimally. It is relatively uninformative to report such extensive gene ontological analysis for muscle proteome data. Many of the tests used to denote enrichment assume a comparison against a whole-genome background and so are not appropriate for muscle proteomic analysis that does not include data all gene products.

Response 2: Our analysis was orientated on two high impact manuscripts, also investigating skeletal muscle fibres in combination with proteome analysis using GO-categorization (Kleinert, 2018; Murgia, 2017. But we performed an additional pathway analysis, with PANTHER to show more detailed information about pathways showing the response to resistance training.

Point 3: Indeed the results sections lacks information regarding the total number of proteins that were included in the current analysis.

Response 3: Because Progenesis only uses comparisons between to samples and quantifies them, we don’t get any information about a raw number of proteins. But we know that 1.377 proteins and protein groups have been identified in our analysis. We added the following sentence (237-239):

Overall comparisons 1 377 proteins or protein groups were identified and quantified in our analysis.

 Point 4: Moreover, summarising proteins by gene ontology provides limited biological insight compared to that which can be achieved by considering the literature and biochemical attributes of proteins, interaction networks and metabolic pathways, for example.

Response 4: Because pathway analysis often contain only 60 % of all mapped proteins, we chose a GO-categorization. But we added a pathway-analysis performed with PANTER (using default settings for Fisher’s exact and FDR) to show more details about the physiological mechanisms.

The manuscript may also be improved by considering the following minor issues.

Point 5: Is the choice of experiment design a limitation? From the data presented, it is not clear whether there were baseline differences across the muscle proteome of the 2 independent groups of participants.Given the biological variability of human participants, a cross-over design may have been more appropriate/ effective in detecting supplementation-specific effects.

Response 5: I totally agree with your comment! A cross over design and an intra-individual comparison between a 12-week training intervention with and without collagen supplementation would have had a lot of advantages. But for economic reasons, we decided for a parallel design. For a crossover design, we would have needed a wash-out phase of at least 12 weeks and another training intervention of 12 weeks. The dropout rate over 9 month would have been much higher, which would have been a big disadvantage.

Point 6: What is the reproducibility of the anthropometric and physiological outcome data? Some of the data in Tables 1 and 2 seem quite broadly distributed. Does your data include lower and higher responders and could the data be illustrated to show the individual responses of participants?

Response 6: The reliability of 1-RM-testing was investigated in trained individuals (ICC: 0.99) (McBride, 2002) and achieves even in untrained and middle aged participants very high correlations values (ICC: 0.97-0.99) (Levinger, 2009). The Inbody770, that we used in our study, has also already been tested for its reliability and received very good results (ICC: Fat Mass (kg) = 0.99; Fat Free Mass (kg) = 1.00) (McLester, 2018).

Because our participants were randomized without being matched according to specific criteria such as strength parameter, the distributions within the groups are unfortunately different. The variances of the exercise SQ are unfortunately not homogeneous, which was taken into account in the statistics, but which results in high standard deviations. We also suspect that we had some high and low responders in the groups, but at the moment we can’t analyse the individual proteome data in connection with the resistance training outcomes, because the absolute quantification via immunoblotting is still missing.

Taking the individual strength developments into account, we can see some high and low responder, but in the collagen group a definitely higher trend for increasing strength in all four parameters:

Point 7: Can more insight be extracted from the data by investigating correlations between the muscle proteome and physiological data?

Response 7: Unfortunately, our data do not provide such an analysis. We don’t get an absolute quantification of the proteins with whom a correlation analysis could be performed. Because the ionization differs between proteins, therefore we can’t inference from the relative abundance to absolute values.

Notes/ short questions:

Point 10: Was the biopsy site standardised? Which of the 2 biopsy samples (pre- or post-warm-up) was used for comparing longitudinal differences in response to 12-weeks training?

Response 10: Muscle biopsy was standardised taken from the vastus lateralis muscle of each participant’s right leg. I changed the text as follows (138-141; 145-148):

Next, a muscle biopsy was taken standardised percutaneously from the vastus lateralis muscle of the right leg under local anesthesia (Xylocitin® 2 % with Epinephrine, mibe GmbH, Brehna, Germany) with a 5-mm Bergstrøm needle, as described previously. … and another muscle biopsy was taken 3 cm proximal to the previous cut from the same leg. The proteome analysis was performed with the muscle sample collected before training in the fasted state to analyze longitudinal differences.

Point 11: Usually necessary to provide a reference number to confirm appropriate ethic review 

Response 11: The investigation was not registered as a clinical trial, but we received a reference number from our ethics commission: EKS V 01/2016. The positive ethics vote can be found in the appendix.

Kleinert, M.; Parker, B.L.; Jensen, T.E.; Raun, S.H.; Pham, P.; Han, X.; James, D.E.; Richter, E.A.; Sylow, L. Quantitative proteomic characterization of cellular pathways associated with altered insulin sensitivity in skeletal muscle following high-fat diet feeding and exercise training. Scientific reports 2018, 8, 10723. 10.1038/s41598-018-28540-5.

Levinger, I.; Goodman, C.; Hare, D.L.; Jerums, G.; Toia, D.; Selig, S. The reliability of the 1RM strength test for untrained middle-aged individuals. Journal of science and medicine in sport 2009, 12, 310–316. 10.1016/j.jsams.2007.10.007.

McBride, J.M.; Triplett-McBride, T.; Davie, A.; Newton, R.U. The effect of heavy- vs. light-load jump squats on the development of strength, power, and speed. Journal of strength and conditioning research 2002, 16, 75–82.

McLester, C.N.; Nickerson, B.S.; Kliszczewicz, B.M.; McLester, J.R. Reliability and Agreement of Various InBody Body Composition Analyzers as Compared to Dual-Energy X-Ray Absorptiometry in Healthy Men and Women. Journal of clinical densitometry : the official journal of the International Society for Clinical Densitometry 2018. 10.1016/j.jocd.2018.10.008.

Murgia, M.; Toniolo, L.; Nagaraj, N.; Ciciliot, S.; Vindigni, V.; Schiaffino, S.; Reggiani, C.; Mann, M. Single Muscle Fiber Proteomics Reveals Fiber-Type-Specific Features of Human Muscle Aging. Cell reports 2017, 19, 2396–2409. 10.1016/j.celrep.2017.05.054.

Reviewer 2 Report

Dear author of "Effects of 12 Weeks of Hypertrophy Resistance Exercise Training Combined with Collagen Peptide Supplementation on the Skeletal Muscle Proteome in Recreationally Active Men"

I carefully read the manuscript and I believe this manuscript is worth publishing on 

Nutrients after minor revise. The manuscript contains interesting information about the effect of collagen peptides on skeletal muscle proteome after resistance training.

On the other hand I have some questions, please provide the comment for my questions.

Q1 The manuscript examined the effect of collagen peptides on skeletal muscle compared with control group. However present results might possibly obtained after ingestion of other protein hydrolysate, such as soy peptides and/or milk peptides not only collagen hydrolysate. Please provide a comment for my question.

Q2 Are there any speculation for active components such as short chain peptide? It has been reported that short chain peptides were detected as food-derived collagen peptide in human blood after ingestion of collagen hydorlysates (peptides). Author mentioned increase of free hydroxyproline (Hyp), however bioactivities of Hyp containing peptide are reported rather than free Hyp. Author should consider active components and Hyp peptides in discussion.

Best regards.

Author Response

Response to Reviewer 2 Comments:

First, I would like to thank you very much for reading my manuscript and for the many constructive comments! I hope that I was answering most of the questions and comments or that I was able to work on them appropriately!

Thank you very much!

Point 1:

My major Q1 The manuscript examined the effect of collagen peptides on skeletal muscle compared with control group. However present results might possibly obtained after ingestion of other protein hydrolysate, such as soy peptides and/or milk peptides not only collagen hydrolysate. Please provide a comment for my question.

Response 1:

Muscle growth is (according to current literature) stimulated significantly stronger by resistance exercise training in combination with protein supplementation than by resistance training alone. The key component in the protein appears to be the essential amino acid leucine, which activates the mTOR pathway leading to muscle hypertrophy.

While whey and soy protein contain 13.6% and 8.0% leucine (van Vliet et al., 2015), collagen has only 2.7% (Zdzieblik et al., 2015).

According to Philips (2016), the muscle protein synthesis after ingestion of collagen would be insufficient to stimulate muscle hypertrophy compared to whey and soy. Nevertheless, our strength and anthropometric results show a higher increase in the collagen group compared with placebo, suggesting that some other mechanisms are responsible for the outcome.

Milk protein leads to higher muscle hypertrophy vs. isonitrogenous soy protein (Hartmann et al., 2007), but elevating the total amount of plant-based protein to match the essential amino acid content, results difference between whey and soy proteins vanished (Review: van Vliet et al., 2015). Both sources activate the mTOR-pathway and downstream key components like 4E-BPI, p70S6K, ERK1/2, Pi3K, Akt (Anthony et al., 2007; Moore et al., 2010) through which hypertrophy seems to occur. Our results show an increase in fat free body mass and higher strength gains in the collagen group without significant changes in abundance in the proteins named above as expected because of the physiological results.

These hypotheses are speculative and derived from studies that investigated single proteins. Because to our knowledge there are no investigations analyzing the skeletal muscle proteome after resistance training in combination with protein supplementations. This could be an interesting approach for further investigations especially comparing different protein sources.

Point 2:

Are there any speculation for active components such as short chain peptide? It has been reported that short chain peptides were detected as food-derived collagen peptide in human blood after ingestion of collagen hydorlysates (peptides). Author mentioned increase of free hydroxyproline (Hyp), however bioactivities of Hyp containing peptide are reported rather than free Hyp. Author should consider active components and Hyp peptides in discussion.

Response 2:

Unfortunately, we did not measure short chain peptides. Free hydroxyproline (Hyp) was only measured to exclude the possibility that all participants could utilize the collagen.

However, studies show similar increases in Hyp-peptides and free Hyp with a ratio approximately 1:3 after collagen hydrolysate ingestion (Iwai et al., 2005). For example, Pro-Hyp, Ala-Hyp-Gly, Ser-Hyp-Gly and Hyp-Gly have been identified after collagen hydrolysate ingestion (Ichikawa et al., 2009; Shigemura et al., 2009, 2011; Sugihara et al., 2012). Whereas Pro-Hyp has been previously described affecting skin and bone health (Shigemura et al., 2009), Kitakaze et al. (2016) showed that the dipeptide Hyp-Gly increased myoblast differentiation and myotube hypertrophy in murine skeletal muscle C2C12 cells when Hyp-Gly concentration was higher than 10.0 µM. Further investigations show Hyp-Gly concentration of 4.2 µM in human plasma followed 8 g collagen hydrolysate ingestion (Sugihara et al., 2012). An appropriate amount to induce hypertrophy through Hyp-Gly seems to occur with approximately 20 g, implying the effect behaves linear. As we ingested 15 g collagen hydrolysate daily, possibly the effect appears slightly weakened, as we can’t show an increase in Myosin Heavy Chain.

Furthermore, Pro-Hyp and Hyp-Gly lead to an increase in fibroblast growth, which are included in the proliferation and differentiation of myogenic progenitor cells (Fry et al., 2017).

Overall, we can’t prove a difference in muscle cell growth between collagen and placebo group, that will be shown in another publication (under review: Kirmse et al., 2019).

Thank you for this comment, I added this topic to the discussion (429-432).

                 References:

Anthony, T.G.; McDaniel, B.J.; Knoll, P.; Bunpo, P.; Paul, G.L.; McNurlan, M.A. Feeding meals containing soy or whey protein after exercise stimulates protein synthesis and translation initiation in the skeletal muscle of male rats. The Journal of nutrition 2007, 137, 357–362. 10.1093/jn/137.2.357.

Fry, C.S.; Kirby, T.J.; Kosmac, K.; McCarthy, J.J.; Peterson, C.A. Myogenic Progenitor Cells Control Extracellular Matrix Production by Fibroblasts during Skeletal Muscle Hypertrophy. Cell stem cell 2017, 20, 56–69. 10.1016/j.stem.2016.09.010.

Hartman, J.W.; Tang, J.E.; Wilkinson, S.B.; Tarnopolsky, M.A.; Lawrence, R.L.; Fullerton, A.V.; Phillips, S.M. Consumption of fat-free fluid milk after resistance exercise promotes greater lean mass accretion than does consumption of soy or carbohydrate in young, novice, male weightlifters. The American journal of clinical nutrition 2007, 86, 373–381. 10.1093/ajcn/86.2.373.

Ichikawa, S.; Morifuji, M.; Ohara, H.; Matsumoto, H.; Takeuchi, Y.; Sato, K. Hydroxyproline-containing dipeptides and tripeptides quantified at high concentration in human blood after oral administration of gelatin hydrolysate. International journal of food sciences and nutrition 2010, 61, 52–60. 10.3109/09637480903257711.

Iwai, K.; Hasegawa, T.; Taguchi, Y.; Morimatsu, F.; Sato, K.; Nakamura, Y.; Higashi, A.; Kido, Y.; Nakabo, Y.; Ohtsuki, K. Identification of food-derived collagen peptides in human blood after oral ingestion of gelatin hydrolysates. Journal of agricultural and food chemistry 2005, 53, 6531–6536. 10.1021/jf050206p.

Kitakaze, T.; Sakamoto, T.; Kitano, T.; Inoue, N.; Sugihara, F.; Harada, N.; Yamaji, R. The collagen derived dipeptide hydroxyprolyl-glycine promotes C2C12 myoblast differentiation and myotube hypertrophy. Biochemical and biophysical research communications 2016, 478, 1292–1297. 10.1016/j.bbrc.2016.08.114.

Moore, D.R.; Atherton, P.J.; Rennie, M.J.; Tarnopolsky, M.A.; Phillips, S.M. Resistance exercise enhances mTOR and MAPK signalling in human muscle over that seen at rest after bolus protein ingestion. Acta physiologica (Oxford, England) 2011, 201, 365–372. 10.1111/j.1748-1716.2010.02187.x.

Phillips, S.M. The impact of protein quality on the promotion of resistance exercise-induced changes in muscle mass. Nutrition & metabolism 2016, 13, 64. 10.1186/s12986-016-0124-8.

Shigemura, Y.; Iwasaki, Y.; Tateno, M.; Suzuki, A.; Kurokawa, M.; Sato, Y.; Sato, K. A Pilot Study for the Detection of Cyclic Prolyl-Hydroxyproline (Pro-Hyp) in Human Blood after Ingestion of Collagen Hydrolysate. Nutrients 2018, 10. 10.3390/nu10101356.

Shigemura, Y.; Iwai, K.; Morimatsu, F.; Iwamoto, T.; Mori, T.; Oda, C.; Taira, T.; Park, E.Y.; Nakamura, Y.; Sato, K. Effect of Prolyl-hydroxyproline (Pro-Hyp), a food-derived collagen peptide in human blood, on growth of fibroblasts from mouse skin. Journal of agricultural and food chemistry 2009, 57, 444–449. 10.1021/jf802785h.

Sugihara, F.; Inoue, N.; Kuwamori, M.; Taniguchi, M. Quantification of hydroxyprolyl-glycine (Hyp-Gly) in human blood after ingestion of collagen hydrolysate. Journal of bioscience and bioengineering 2012, 113, 202–203. 10.1016/j.jbiosc.2011.09.024.

van Vliet, S.; Burd, N.A.; van Loon, L.J.C. The Skeletal Muscle Anabolic Response to Plant- versus Animal-Based Protein Consumption. The Journal of nutrition 2015, 145, 1981–1991. 10.3945/jn.114.204305.

Zdzieblik, D.; Oesser, S.; Baumstark, M.W.; Gollhofer, A.; Konig, D. Collagen peptide supplementation in combination with resistance training improves body composition and increases muscle strength in elderly sarcopenic men: a randomised controlled trial. The British journal of nutrition 2015, 114, 1237–1245. 10.1017/S000711451500281

Round  2

Reviewer 1 Report

I thank the authors for their responses to my comments. The manuscript is improved but the authors have not fully addressed my main issues regarding statistical analysis and reporting of data.

Author Response

I thank the authors for their responses to my comments. The manuscript is improved but the authors have not fully addressed my main issues regarding statistical analysis and reporting of data.

Response:

I am not sure what your main issue is regarding our statistical analysis. With our present data set we can only calculate mean and sd of the normalized abundance, but we decided to use fold change and FDR-values, because this is more common in publications describing proteome data. We think this would mean no further information. We could also calculate the CV for the whole data set, but that would only give a description of the variation of the total data and not about the proteins we selected. Nevertheless, immunoblots will be performed soon for further statistical analysis.

Changes in the second round are highlighted in blue.

Looking at the entire dataset, we note that we excluded some participants from our anthropometric and statistical analysis. We are very sorry! Therefore, the anthropometric and strength values differ slightly from the latest version. But fortunately there were only small changes in the results.

Regarding your note to the conclusions, we distribute the parts, which are a kind of discussion, to the corresponding result chapter. The conclusion has now become more concrete and shorter. We have also included future directions in our conclusions and we are hoping it'll do better!